# Inhibition of Caspase-2 Translation by the mRNA Binding Protein HuR: A Novel Path of Therapy Resistance in Colon Carcinoma Cells?

**DOI:** 10.3390/cells8080797

**Published:** 2019-07-30

**Authors:** Wolfgang Eberhardt, Usman Nasrullah, Kristina Haeussler

**Affiliations:** Institut für Allgemeine Pharmakologie und Toxikologie, Pharmazentrum Frankfurt/ZAFES, Universitätsklinikum und Goethe-Universität, Theodor-Stern-Kai 7, D-60590 Frankfurt am Main, Germany

**Keywords:** colorectal cancer, caspase-2, cell survival mechanisms, DNA damage response, human antigen R (HuR), internal ribosomal entry site (IRES), RNA binding proteins

## Abstract

An increased expression and cytoplasmic abundance of the ubiquitous RNA binding protein human antigen R (HuR) is critically implicated in the dysregulated control of post-transcriptional gene expression during colorectal cancer development and is frequently associated with a high grade of malignancy and therapy resistance. Regardless of the fact that HuR elicits a broad cell survival program by increasing the stability of mRNAs coding for prominent anti-apoptotic factors, recent data suggest that HuR is critically involved in the regulation of translation, particularly, in the internal ribosome entry site (IRES) controlled translation of cell death regulatory proteins. Accordingly, data from human colon carcinoma cells revealed that HuR maintains constitutively reduced protein and activity levels of caspase-2 through negative interference with IRES-mediated translation. This review covers recent advances in the understanding of mechanisms underlying HuR’s modulatory activity on IRES-triggered translation. With respect to the unique regulatory features of caspase-2 and its multiple roles (e.g., in DNA-damage-induced apoptosis, cell cycle regulation and maintenance of genomic stability), the pathophysiological consequences of negative caspase-2 regulation by HuR and its impact on therapy resistance of colorectal cancers will be discussed in detail. The negative HuR-caspase-2 axis may offer a novel target for tumor sensitizing therapies.

## 1. Introduction

Colorectal cancer (CRC) is one of the most common cancers in the western world. Despite having a slow progression, it is characterized by the high tumor mortality that is mainly caused by the strong metastatic potential of the primary tumor [1,2]. Pathologically, in the majority of CRC cases, tumors appear spontaneously with a number of risk factors essentially contributing to the sporadic CRC. Hereby, age, adverse dietary habits, as well as a variety of diseases including diabetes, obesity, and chronic inflammatory diseases, namely, those which affect the small and/or large intestine such as Crohn´s disease or ulcerative colitis, strongly increase the risk of CRC [1,3,4]. Furthermore, inherited mutations underlying familial adenomatous polyposis (FAP) and mismatch mutations due to defective DNA repair are important risk factors which predispose individuals to the development of CRC (for a review see [5]). At the molecular level, germline mutations in the tumor suppressor gene, adenomatous polyposis coli (Apc), which encodes a cytoplasmic protein that, amongst others, binds to **β**-catenin thereby impairing its capability to activate the Wnt signaling pathway, seem to be critical for increased proliferation of epithelial tumor cells. However, mutations in Apc are not exclusively found in FAP patients but also in many sporadic colorectal tumors (for a review see [6,7]). Data from target prediction and pathway analysis implicate genetic mutations which together with external risk factors affect critical signaling pathways that are essential for keeping the homeostasis of a rapidly self-renewing tissue and for maintaining the epithelial integrity of the intestine. Prominent examples include the mitogen activated protein kinase (MAPK), Wnt/β-catenin, Notch, p53, phosphoinositide-3-(PI3) kinase, and transforming growth factor (TGF) β signaling pathways (for a review see [5,8]). Alterations in these signaling cascades play a pivotal role in the process of colonic epithelial transformation and in metastasis of CRC. Over the last decade, an increasing body of evidence has revealed that besides genetic and epigenetic events, changes in the post-transcriptome encompassing mechanisms that can affect different phases of RNA maturation, including trafficking, degradation and translation, critically contribute to CRC development (for a review see [9,10]). Functionally, these mechanisms allow for a rapid adaptation to external stress conditions which frequently lead to a global repression of transcription. Post-transcriptional regulation of a given mRNA is principally controlled by the occupancy of specific cis-regulatory elements with at least two types of transacting factors including the non-coding RNAs, namely microRNAs (miRs) and the so-called turnover and translation regulatory (TTR) RNA binding proteins (RBPs), in other words, TTR-RBPs. Importantly, TTR-RBPs and miRs bind adjacent cis-regulatory elements and thereby control shared target mRNAs in a cooperative or an antagonistic manner [11]. In particular, the TTR-RBPs with a high binding affinity to adenylate (A) and uridylate (U) rich elements (AREs) or U-rich sequences most frequently found in the 3´untranslated region (UTR), but also in the coding region or 5´UTR of target mRNAs, are amongst the best characterized RBPs (for a review see [12,13]). Currently, more than 20 ARE-associated RBPs with a strong regulatory impact on mRNA stability and/or translation are known [13]. According to current estimations, approximately 8% of the human transcripts contain one or more AREs in their UTRs, many of these mRNAs encoding tumor-related proteins (for a review see [14,15,16]). There is solid rationale to suggest that ARE-dependent post-transcriptional gene regulation plays a key role in the initiation and progression of CRC. For an in-depth overview of the role of different ARE-RBPs in CRC we refer to an up-to-date review by our colleagues [10].

## 2. The Role of Human Antigen R (HuR) in Colon Carcinogenesis

One of most intensively studied ARE-RBPs is the ubiquitous human antigen R (HuR), which is a member of the embryonic lethal abnormal vision (ELAV) family proteins (for reviews see [15,16,17,18,19]). Data from many different laboratories demonstrate that HuR is a multitasking factor which is involved in almost every aspect of mRNA maturation and processing. Using high throughput HuR-immunoprecipitation-based approaches (RIP-CHIP, PAR-CLIP), the number of novel putative mRNA targets of HuR has greatly expanded during the last years [20,21]. HuR can affect the expression of many key regulatory factors of proliferation, cell cycle, migration, angiogenesis, metastasis, inflammation, survival, and senescence [15,17,18]. For this reason, HuR is considered a promising target for novel anticancer therapy (for an up-to-date review see [22]). The strong tumorigenic impact of HuR is further underscored by the fact that it is upregulated in a vast majority of human cancers including CRC [23,24,25,26,27,28,29,30]. In accordance, high expression levels of HuR tightly correlate with tumor grade and malignancy and, consequently, also with a worse prognosis [31] (for a review see [32]). A similar correlation was also observed in sporadic colorectal cancer [33] and in patients with FAP [34]. Interestingly, a recent study that employed the HuR small molecule inhibitor MS-444 revealed a critical role of HuR in the early onset of FAP but not in inflammatory bowel disease, thus, pointing to a cell-type specific impact of HuR on intestinal physiology and pathology [35]. Up to now, to the best of our knowledge, investigations to explain the increased expression of HuR in tumors have not described mutations in the promoter region or in the coding region of the HuR gene leading to a malfunction of HuR. Therefore, deregulations in HuR seem mainly attributed to altered signaling pathways controlling the activity of HuR. Since HuR is predominantly localized in the nucleus, its presence in the cytoplasm is intimately linked to its mRNA stabilizing and translation modulatory functions [36,37]. Consequently, an important feature of pathologically increased post-transcriptional activity of HuR is an alteration of its subcellular localization, probably due to impaired HuR post-translational modifications (for a review see [38]). Accordingly, investigations of paired human normal and colon cancer tissues have illustrated a strong increase mainly in cytoplasmic HuR staining as compared with normal tissue [33]. Elevated cytoplasmic HuR levels were also found in the colonic epithelial cells of patients with inflammatory bowel disease and associated adenoma [35]. Noteworthy, HuR plays indispensable roles in embryonic development [39] and various physiological processes including the maturation of hematopoietic progenitor cells [40]. The same study demonstrated critical roles of HuR in intestinal epithelial cell growth and differentiation [40]. The authors of this study, furthermore, demonstrated that the proliferative and anti-apoptotic activity of HuR is critical for tissue architecture of the gastrointestinal tract. Consequently, global postnatal deletion of HuR is accompanied by an extensive loss of microvilli and leads to severe cachexia [40]. Some prominent HuR targets which are critically involved in the growth of the intestinal mucosa include the Wnt coreceptor LDL receptor-related protein 6 (LRP6) [41], the proto oncogene c-Myc [42], and cyclooxygenase-2 (COX-2) [30,43]. The role of different RNA binding proteins for gastrointestinal epithelial homeostasis is comprehensively reviewed in previous publications by our colleagues [10,11].

### 2.1. HuR: A Unique Post-Transcriptional Regulator of Cell Survival

Pathologically, evasion of apoptosis is a hallmark of most tumor cells. In addition, the impaired sensitivity of tumor cells towards apoptotic stimuli activating either receptor mediated extrinsic or mitochondrial-mediated intrinsic apoptotic cascades is an important feature of therapy resistance of tumor cells toward chemotherapeutic drugs and/or γ-irradiation [14,17,18,19]. HuR can directly promote drug resistance through post-transcriptional regulation of proteins of the ATP binding cassette (ABC) transporter family, including the P-glycoprotein (encoded by the ABCB1 gene) [31] and the ATP subfamily G member 2 (ABCG2) [44]. Accordingly, an increased expression of ABC transporters was demonstrated in various drug-resistant cell lines suggesting overexpression of these proteins plays an important role in the development of multi-drug resistance (for a review see [45]). Notably, high levels of ABCG2 were found mainly in the tissue from colorectal carcinoma patients with a high resistance to 5-fluorouracil-based chemotherapy and, importantly, the same tissues revealed an increased expression of HuR but highly reduced levels of mir-519 [44], one of the well-characterized HuR suppressing miRs in colon carcinoma and other human tumor cells [46]. From their data, it is tempting to speculate that HuR via increasing ABCG2 expression, represents a potential target for sensitizing CRC patients to chemotherapy. In most mammalian cells, HuR does exert a broad post-transcriptional cell survival program [14], mainly through stabilizing and/or enhancing the translation of mRNAs coding for anti-apoptotic effector proteins including prothymosin α [47], B-cell CLL/lymphoma 2 protein (Bcl-2) [48], p21 [49], Mcl-1 [14], X-linked inhibitor of apoptosis (XIAP) [50], survivin [51], and the deacetylase stress-response and chromatin-silencing factor Sirtuin 1 (SIRT1) [52] (reviewed in [53]). Conversely, HuR can impair the translation of pro-apoptotic factors like p27 [54], c-Myc [55], and the type 1 insulin growth factor-receptor (IGF-IR) [56]. In accordance with these reports, we previously identified the pro-apoptotic caspase-2L (also called ICH-1L) as a novel target gene, in colon carcinoma cells, which is negatively regulated by HuR, mainly through an inhibition of translation [57]. In contrast to other members of the caspase family, the defined regulatory role of caspase-2 in the apoptotic process is still debatable (for reviews see [58,59,60]). Nonetheless, data from various studies indicate that caspase-2, by acting as an apical caspase, is critically involved in apoptosis induced by DNA damage [61,62,63]. Interestingly, in addition to DNA damage-induced apoptosis caspase-2 exerts various important non-apoptotic functions including tumor suppression [64,65,66], maintenance of genome surveillance by controlling cell cycle checkpoints [67,68], and autophagy [69]. Data from various tumor models implicate that loss of caspase-2 facilitates tumor growth and contributes to the development of cancer [64,65,66] (for a review see [70]). Correspondingly, a decrease in caspase-2 expression was evidently shown in several types of human cancers [71,72,73], thus, indicating a tumor suppressive role of caspase-2. Whereas research investigations of the last decade have mainly focused on the elucidation of mechanisms of casapse-2 activation, only a few studies have investigated mechanisms which control caspase-2 expression. Some of these studies showed that caspase-2 expression is critically controlled by miR-dependent repression [74,75,76]. In addition, a recent publication demonstrated that transcription of caspase-2 mRNA in colon carcinoma cells is controlled by B-cell CLL/lymphoma 9 protein (BCL9L) [68]. A dysfunction of this pathway critically contributes to the pathological tolerance towards aneuploid tumor cells in human colorectal carcinoma [68]. Moreover, caspase-2 is a target of alternative splicing which gives rise to two proteins with antagonistic functions, and thus suggesting that caspase-2 can exert dual roles in cell death regulation [77]. A high expression of a second splice variant of caspase-2 mRNA generating the truncated protein caspase-2S with an anti-apoptotic activity was observed in macrophages derived from human atherosclerotic plaques [78]. However, to the best of our knowledge, functional relevance of this splicing variant in vivo has not been demonstrated so far. Interestingly, topoisomerase inhibitors I or II were shown to trigger a switch between caspase-2L and caspase-2S mRNA species via exon inclusion [79,80]. Considering that several reports have demonstrates a critical role of HuR in mRNA splicing [81,82,83], the possible impact of HuR on a chemotherapy-induced switch between the pro-and anti-apoptotic caspase-2 splice variants is an interesting issue which warrants further investigation.

### 2.2. Negative Caspase-2 Regulation by HuR as a Novel Survival Mechanism in Adenocarcinoma Cells

Most data from the last decade clearly implicate that the post-transcriptional cell survival program by HuR is highly relevant for the protection of tumor cells from harmful external stimuli. These include inducers of DNA damage response (DDR), a specific cell signaling program which coordinates cell cycle checkpoints, DNA repair, and apoptosis in order to prevent and delay genetic instability and tumorigenesis [84]. The identification of caspase-2L as an additional target of HuR in different human colon carcinoma cells [57] is in alignment with reports demonstrating a negative influence on protein translation by HuR. Thereby, the modulatory effects on translation are mediated by a direct interaction of HuR with the 5´UTR of respective mRNAs including p27 [54,85], IGF-IR [56], and thrombomodulin [86], although none of these reports reference observations from intestinal epithelial cells. Common to all these cases, the translational inhibition is due to an interference of HuR with an internal ribosome entry site (IRES) driven translation of related mRNAs. Importantly, although the 3´UTR of caspase-2 bears a couple of prototypical HuR binding sites, an interaction of HuR with these elements could not be confirmed by RNA-crosslinking or by biotin pull-down experiments, thus, implying a preferential binding of HuR to the 5´UTR of caspase-2 [57]. Functionally, the transient knockdown of HuR significantly increased the sensitivity of colon carcinoma cells to γ-irradiation or drug-induced apoptosis and, importantly, the sensitizing effects were fully rescued after additional silencing of caspase-2 [87,88]. Since the caspase-2-mediated sensitization was observed in p53-mutated, as well as in wildtype p53 cells, it is tempting to suggest that the effect on apoptosis occur independent of the p53 status. Thus, targeting the constitutive binding of HuR to caspase-2 mRNA may represent a promising therapeutic approach for sensitizing colorectal carcinoma cells to currently used anti-tumor therapies.

### 2.3. HuR Affects Internal Ribosome Entry Site (IRES) Mediated Caspase-2 Translation

IRES-mediated translation represents an alternative mode of translation which allows cells to bypass the energy-consuming process of scanning a 48S ribosomal complex along the mRNA underlying cap-dependent translation. A large body of evidence has emerged indicating IRES dependent translation in eukaryotic cells is mainly utilized under physiological or pathological stress conditions when cap-dependent initiation is severely compromised including viral infection, hypoxia, apoptosis, or DNA damage (for reviews see [89,90,91]). In addition to its well-established mRNA stabilizing functions on ARE bearing mRNAs, HuR has been identified as a negative modulator of IRES-dependent translation [54,56,92]. Mechanistically, HuR can interfere with IRES-mediated translation by arresting the IRES-associated pre-initiation complex in an inactive state as previously shown for the IRES of the IGF-IR [56]. Similar to this study, the inhibition of the cyclin dependent kinase (Cdk) inhibitor p27^Kip1^ is attributed to a direct binding of HuR to an IRES in close proximity to the p27 initiation codon [54]. By mapping the critical IRESs, the authors identified several U-rich elements [54] which, in addition to the AREs, represent prototypical HuR RNA binding sites [13,93]. Functionally, p27^Kip1^ limits cell growth via the induction of cell cycle arrest. Therefore, downregulation of p27, which is observed in many tumor samples including those from colorectal cancer as a consequence of a loss of the translation regulatory protein poly C binding protein 1 (PCBP1), underlines the pathophysiological significance of p27 deficiency in tumorigenesis [94]. Importantly, besides reducing translation of p27, HuR can increase the expression of cyclin A and B through cell cycle-dependent stabilization of the respective mRNAs [95]. Thus, overexpression of HuR can promote cell cycle progression at different levels, a feature which may gain specific importance in tissues with a high turnover rate.

The list of IRES-controlled mRNAs is steadily growing and includes capped mRNAs with functional IRES elements in their 5´UTRs and which are refractory to translational repression as they can switch from cap to IRES-dependent translation (for a review see [91]). Thus, it is not surprising that several key players in apoptosis and cell survival are regulated through IRES-dependent translation. Prominent examples include both pro-apoptotic factors such as c-Myc [96] and a protease-activating factor 1 (Apaf-1) [97], and also apoptosis inhibitory proteins such as XIAP [98], Bcl-2 [99], cellular inhibitor of apoptosis protein 1 (cIAP1) [100], and p27 [54]. In this respect, caspase-2 can easily join the list of IRES-controlled target mRNAs. The structural details of the IRES of caspase-2 which is specifically targeted by HuR are still unknown. We cannot rule out that the HuR-mediated block of caspase-2 translation is additionally due to a HuR-dependent sequestration of caspase-2 mRNA to stress granules or processing (P) bodies, both representing ribonucleoprotein (RNP) granules which are typically formed for storage of translationally-arrested mRNAs under stress conditions [101]. However, since our data from loss-of-function experiments suggest a constitutive blockade of IRES translation under normal cell culture conditions, it is rather unlikely that this mechanism is operative in colon carcinoma cells.

### 2.4. HuR as a Potential Negative IRES Trans-Acting Factor (ITAF) of Caspase-2 Translation

The binding of IRES by several canonical initiation factors together with accessory proteins, termed “IRES trans-acting factors” (ITAFs), is a major prerequisite for initiation of IRES-mediated translation (for reviews see [91,92,102]). Mechanistically, ITAFs in coordination with translation initiation factors and other RBPs facilitate the RNA binding to ribosomal 40S subunits. Alternatively, ITAFs function as RNA chaperons which by favoring a certain secondary structure promote direct binding of the the 40S ribosome subunit to the mRNA [89]. Some of the currently identified ITAFs include nucleo-cytoplasmic shuttling proteins such as heterogeneous ribonucleoproteins (hnRNPs) RNP1 (synonymously called pyrimidine-tract binding protein, i.e., PTB), hnRNPC1 and C2, p54nrb, programmed cell death protein 4 (PDCD4), nucleolin, and HuR (for reviews see [91,103]). It is important to note that IRES-dependent translation is regulated by multiple RBPs rather than by one single ITAF. Most of these RBPs are implied in the control of different aspects of mRNA metabolism, for example, splicing, transcription, turnover and ribosome biogenesis [103]. Since many of the known ITAFs are predominantly localized in the nucleus, the nucleo-cytoplasmic shuttling represents a frequent mode of regulating ITAF activity. However, it is worth mentioning that IRES binding is not obligatorily restricted to the cytoplasm but already occurs in the nucleus, and thereby ensures that IRES-containing mRNA remain separated from the translational machinery [104,105]. Additional modes of ITAF regulation include different post-translational modifications including protein kinase-dependent phosphorylation (e.g., the regulation of c-Myc and cylin D IRESs through Akt-dependent phosphorylation of hnRNPA1 [106]). In accordance, HuR constitutes another well documented example of an RBP which is a target of diverse post-translational modifications, namely, protein kinase-triggered phosphorylation (for a review see [38]). Although the impact of these post-translational HuR modifications on the IRES-dependent translation has not been addressed so far, it seems plausible that they could also be relevant for modulation of its ITAF activity. Our previous finding that protein kinase C (PKC) triggered phosphorylation of different HuR domains coordinate RNA binding affinity to ARE-bearing mRNAs and subcellular localization of HuR offers an elegant mechanism how different ITAF activities could be coordinated as well. Notably, the constitutive HuR phosphorylation at serine 318 by PKCδ observed in human colon carcinoma cells is functionally relevant for the increased HuR binding to target mRNA and, subsequently, for the increased expression of some tumor relevant genes including cyclooxygenase-2 and cyclins [107]. Interestingly, pharmacological inhibition of PKC activity, in addition to reducing HuR phosphorylation at serine 318, had a strong inhibitory effect on the high cytoplasmic HuR abundance [107], thus, implying the high constitutive HuR phosphorylation may be causative for the pathologically-increased cytoplasmic HuR levels observed in tissue specimens from CRC patients or patients with early adenomas or inflammatory bowel diseases [10]. The pathophysiological relevance of this particular post-translational HuR modification is, furthermore, implied by data from immunohistochemistry demonstrating a robust increase in HuR-phosphorylation at serine 318 as compared with colon tissue from healthy donors [107]. Importantly, a previous publication demonstrated that PKC promotes IRES-dependent translation of the mRNA coding for the transcription factor GATA4 in response to vasopressin [108]. Therefore, it would be intriguing to investigate whether the constitutive PKCδ-induced phosphorylation of HuR in colon carcinoma cells may have a critical impact on HuR binding to IRES located in the 5´UTR of caspase-2 as well (Figure 1). In addition to different PKC isoenzymes, HuR is a direct target of the checkpoint kinase 2 (Chk2) [52] and cyclin dependent kinase 1 (Cdk1) [109] with both kinases being pivotally involved in the DDR. Furthermore, all of these protein kinases have a critical impact either on nucleo-cytoplasmic HuR distribution (Cdk1) or on the RNA binding affinity (Chk2) or, as in the case of PKCs, can modulate both HuR modalities simultaneously [110,111]. Data from human HCT116 colorectal carcinoma cells revealed that Chk2-mediated phosphorylation of HuR at serine 88 and 100 and at threonine 118 facilitates the dissociation of HuR from mRNAs coding for proteins with a key role in cell proliferation and apoptosis including the tight junction protein 1, p53 binding protein mouse double minute 2 (Mdm2), K-RAS, SOX4, SIRT 1, and the Bcl-2-associated X protein (BAX) [112]. Importantly, this process is triggered by ionizing radiation (IR) suggesting that the release of HuR-bound mRNA via the Chk2-HuR regulatory axis improves cell survival upon DNA damage [112]. In contrast, polyamines were shown to enhance HuR association to c-Myc mRNA [113] and occludin mRNA [114] via Chk2 dependent HuR phosphorylation, and thereby promote the translation of the corresponding proteins. The seemingly contradictory results from different studies may imply that the final impact of a specific post-translational HuR modification on mRNA binding is primarily controlled by the set of target mRNAs and not by the modification. In light of these observations, we speculate in addition to the constitutive phosphorylation of HuR by PKCδ, the activation of Chk2 in response to genotoxic stress could further enhance the constitutive HuR binding to the caspase-2 mRNA (Figure 1). However, further investigations are needed to define the functional role of specific post-translational HuR modifications by different DNA damage kinases for inhibition of caspase-2. In contrast to this scenario, under conditions of irreparable DNA damage, cytoplasmic HuR itself can switch from a cell survival to a pro-apoptotic factor via apoptosis-induced cleavage [115]. Therefore, another question arising is whether severe DNA damage can induce a dissociation of HuR from caspase-2 mRNA which in turn would promote caspase-2-triggered cell death pathways.

## 3. Regulation of the DNA Damage Response by the HuR-Caspase-2 Axis

### 3.1. HuR and the DNA Damage Response

The DDR has evolved to prevent genetic instability and to avoid replication of damaged DNA mainly though the activation of specific kinase-mediated pathways which coordinate cell cycle arrest, DNA repair and apoptosis (for reviews see [84,116]). Pathologically, a deregulation of the DDR is frequently observed in various types of human malignancies including colorectal cancer [116,117,118]. Tumor cells are endowed with the capacity to escape from DDR-induced cell death which is considered an important step towards malignant transformation and therapeutic resistance [116]. The ataxia telangiectasia mutated (ATM) and the ATM-Rad3-related (ATR) kinases represent the major DNA damage sensing kinases which activate proximal kinases central to the entire DDR. Since DNA damaging conditions do severely repress transcription and translation, it is not surprising that several of the HuR target genes that have been identified so far are also implicated in the DDR induced by genotoxic stress (for a review see [119]). Thus, the close interplay between HuR and the DDR pathway constitutes an important mechanistic feature of drug resistance towards cytotoxic therapies. A prominent example is the HuR-dependent mRNA stabilization of the Cdk inhibitor p21^WAF1^ (synonymously called p21^Cip−1^) which seems critical for G1/S cell cycle arrest upon DNA damage [49]. Since p21 additionally prevents genotoxic stress-induced apoptosis, the HuR-dependent increase in p21 mRNA stability induced by radiotherapy is assumed to confer increased therapy resistance to colon cancer cells [49] (Figure 2A). Another important HuR target which is implied in the DDR is the histone deacetylase SIRT1 which itself exerts pleiotropic prosurvival functions [53]. In turn, in addition to histone modifications, SIRT1 dependent deacetylation is implicated in the modulation and activity of a variety of effectors which promote cell survival and DNA repair including p53, nuclear factor κB (NFκB), the DNA helicase subunit KU70, members of the forkhead family of transcription factors, and BCL-6. For an in-depth review of SIRT1 and DDR we refer to the review article by our colleagues [53]. Of note, a pathological relevance of SIRT1 expression in colorectal carcinoma has been reported by previous studies [120,121,122]. An additional well-known HuR target contributing to cell survival and drug resistance is Mdm2, an E3 ligase which through induction of proteasomal p53 degradation inhibits p53-triggered apoptosis [123,124]. Furthermore, Mdm2 and p53 are controlled by a negative feedback loop, in which p53 induces the transcription of Mdm2, which in turn promotes the degradation of the p53 protein [125] (Figure 2B). Vice versa, Mdm2 can inhibit the transcriptional activity of p53 through directly binding to its N-terminal transactivation domain [126]. Clearly opposite to this, Mdm2 was found to promote p53 synthesis by binding to its mRNA. Thereby, ATM-dependent phosphorylation of Mdm2 following DNA damage is required for the p53 mRNA–Mdm2 interaction in the nucleus and switches Mdm2 from a negative to a positive regulator of p53 [127]. Importantly, both scenarios seem non-mutually exclusive since under conditions of an active ATM pathway, Mdm2 may also act as a positive regulator of p53 while it rapidly degrades p53 when the DNA damage response is switched off. Physiologically, the anti-apoptotic activity by Mdm2 seems highly relevant for the maintenance of the intestinal epithelial architecture since a decrease in Mdm2 expression as a direct consequence of HuR knockout is linked with a strong increase in intestinal epithelial cell apoptosis and a dramatic loss of intestinal microvilli [40]. Since Mdm2 is overexpressed in colon carcinoma and other human cancers, targeting the Mdm2-p53 pathway could offer another promising target for sensitization of various human cancers to chemotherapeutic drugs [128]. Notably, HuR itself is a substrate of Mdm2-triggered neddylation at distinct lysine residues [129]. Increased HuR levels were shown to tightly correlate with Mdm2 abundance in colon cancer metastasis and human hepatocellular carcinoma indicating that this post-translational modification is functionally relevant for malignant transformation of epithelial cells from the liver and the gut [129].

Together these data demonstrate that HuR is not only a post-transcriptional regulator of many DDR-related genes, but also a bona fide target of the DDR. As mentioned before, the activation of DDR kinases jointly influence HuR functions mainly by inducing a change in cytoplasmic HuR abundance and/or by affecting its binding affinity to target mRNA [119]. In particular, the checkpoint kinases, Chk1 and Chk2, which are the direct downstream effectors of the ATM- and ATR-related kinases, both control the binding affinity of HuR to target mRNA either via direct phosphorylation as in the case of Chk2 [112] or, indirectly, via inhibition of Cdk1 which itself induces nuclear retention of HuR to the cytoplasm through direct HuR phosphorylation [109]. Accordingly, lowering HuR levels or preventing its phosphorylation was found to reduce cell survival following genotoxic damage [47,52] which again supports the notion that HuR itself represents a key modulator of the DDR program.

The complex interplay of HuR with the DDR and with DDR-induced cell survival is further emphasized by the identification of caspase-2 as a novel target of HuR [57]. Functionally, the HuR dependent inhibition of caspase-2 translation could, furthermore, constitute an important rescue mechanism of adenocarcinoma cells protecting tumor cells from DNA damage-induced apoptosis. The constitutive binding of HuR to caspase-2 mRNA which we observed in colon carcinoma cells suggests a stable rather than a transient repression of caspase-2 translation, which is further augmented after exposure of cells with either chemotherapeutic agents [88] or γ-irradiation [87] (Figure 1).

### 3.2. Role of Caspase-2 in DNA Damage-Induced Apoptosis

Although caspase-2 null mice are viable and display limited, tissue-specific cell death defects [130], more recent observations with knockout animals revealed an increased susceptibility to tumor formation induced by overexpression of specific oncogenes, suggesting a tumor suppressive role of caspase-2 [64] (reviewed in [131]). In addition, mouse embryonic fibroblasts (MEFs) from these mice are highly resistant to apoptosis induction after cytoskeletal disruption [132]. Caspase-2 deficiency, furthermore, promotes an aberrant DNA damage response and genetic instability after irradiation [63]. In accordance, a growing list of studies support the notion that caspase-2 represents a unifying effector of stress-induced and p53-mediated cell death [133,134] (reviewed in [58]). In addition, caspase-2 is implicated in p53-independent apoptotic responses to DNA damage, particularly under Chk1 compromised conditions. Thereby, caspase-2 is able to sensitize colon carcinoma and other human cancer cell lines to γ-irradiation-induced apoptosis primarily through an evolutionarily highly conserved process which is distinct from mitochondrial and death-receptor-triggered cell death pathways [62]. Furthermore, caspase-2 is highly relevant for deleting mitotically aberrant cells [67]. These studies indicate that caspase-2 acts as a guardian which deletes damaged cells mainly by affecting the cell cycle and DNA damage response (reviewed in [70]). Conversely, under physiological conditions, interdomain phosphorylation of caspase-2 on Ser340 by the mitosis-promoting kinase complex of Cdk-1 and cyclin B1 was found to restrain apoptosis in response to mitotic insults by preventing its enzymatic activation [135]. In addition, evidence is accumulating that caspase-2 is critically involved in the regulation of non-apoptotic functions including cell cycle checkpoints, oxidative stress [136], autophagy [69], and in the maintenance of genomic stability [63,67,137] (Figure 3). The latter aspect will be discussed in more detail in the next paragraph. Consequently, a malfunction of caspase-2 or attenuation of caspase-2 synthesis due to a HuR-mediated repression of translation may jointly impair all of these biological processes as well (Figure 3).

Data from our group similarly demonstrate that, in adenoma carcinoma cells, caspase-2 acts as an apical caspase and as a sensor of drug-induced cell death signals which is constitutively suppressed by HuR [87,88]. Conversely, HuR depletion via increasing caspase-2 levels caused a significant raise in DNA damage-induced apoptosis as indicated by an increase in histone variant H2AX phosphorylation on serine 139 (termed γH2AX) [87]. Phosphorylation of H2AX which is triggered by ATM is one of the earliest events of the DDR at the double-strand break site and represents a valid prognostic factor which is also useful for assessment of responses to therapy of various types of human malignancies including colorectal cancer [138] (reviewed in [117]). Consistent with our data, a downregulation of pro-apoptotic caspase-2 is responsible for malignant transformation of intestinal epithelial cells by the RAS oncogene implying that a reduction of caspase-2 plays a critical role for malignant transformation of the intestine [139]. Interestingly, some important genes which are pivotally involved in the DDR, cell cycle control and survival are shared targets of HuR and caspase-2 which further underlines a close interrelationship of both proteins (Figure 2). Again, a prominent example is the p21^WAF1^ protein which via inhibition of various Cdk-cyclin complexes is critically involved in the p53-induced cell cycle arrest in response to DNA damage [140]. As mentioned before, p21 mRNA is a target of HuR-mediated stabilization [141] (Figure 2A). In addition, p21^WAF1^ is necessary for p53-dependent repression of caspase-2 expression in different human colon carcinoma cell lines [142] (Figure 2A). Strikingly, p21 knockout cells displayed an upregulation of caspase-2 levels above basal levels indicating a constitutive downregulation of caspase-2 by p21^WAF1^ even in the absence of DNA damage [142]. Furthermore, the fact that depletion of the transcription factor, E2F1, induced transcription of caspase-2 may support a model in which E2F1 does mainly act as a repressor of the caspase-2 gene [142]. The negative regulation of caspase-2 transcription by p21^WAF1^ may act in concert with the HuR-triggered repression of caspase-2 translation (Figure 2A). It indicates that HuR limits caspase-2 expression through a direct inhibition of caspase-2 translation and, indirectly, via increasing p21 mRNA stability [141] (Figure 2A). More confounding is the fact that caspase-2 itself was uncovered as an essential cofactor of DNA-damage-induced translation of p21, independent of its enzymatic activity and without the contribution of any of the known caspase-2 activating platforms [143]. Together, these data implicate that p21 translation by caspase-2 is restricted by a negative feedback loop that limits caspase-2 expression (Figure 2A). Importantly, since p21, in addition its original described role in mediating p53-induced cell cycle arrest, is involved in the regulation of various other biological processes including DNA repair, senescence, and apoptosis, the caspase-2 and p21 interplay is likely to critically participate in the control of these processes as well.

The close regulatory interconnection between HuR and caspase-2 in the DNA-damage-induced apoptosis is, furthermore, exemplified by the HuR target Mdm2. As mentioned above, Mdm2 is both, a target and post-translational modifier of HuR, thereby establishing a positive feedback loop which protects HuR from cytoplasmic decay [129]. Interestingly, Mdm2 was previously established as a target of caspase-2 mediated cleavage leading to an inhibition of Mdm2-triggered ubiquitination and, consequently, to the stabilization of p53 [144] (Figure 2B). Thereby, caspase-2 activation in response to DNA damage depends on the PIDDosome, a protein scaffold composed of the p53-inducible protein with a death domain (PIDD) and RIP-associated ICH-I with a death domain (RAIDD) which acts as a major activating platform for caspase-2, namely, in response to genotoxic stressors [145]. Since PIDD itself is a target of p53-induced gene expression, the indirect stabilization of p53 by caspase-2 will ultimately reinforce p53 levels in a positive feedback loop [144] (Figure 2B). Strikingly, a second important caspase-2 activating platform which is built up by a cytosolic complex with receptor-interacting protein kinase (RIP) and the TNF-receptor-associated factor 2 (TRAF-2), is critical for activation of prosurvival pathways executed by the ubiquitous transcription factor, NFκB [146]. A previous study has identified a novel PIDDosome-dependent activation platform of caspase-2 which is confined in nucleoli and specifically induced by genotoxic agents [147]. Consistent with our findings in colon carcinoma cells, earlier studies reported that activation of caspase-2 is critical for an activation of apoptosis in response to doxorubicin [148] or to taxanes [149]. Mechanistically, the increase in intrinsic apoptosis via cleavage of the pro-apoptotic protein Bid and release of cytochrome C is most probably exerted by mitochondrial caspase-2. These data support the notion that the subcellular localization is important for regulation of apoptosis by this caspase [150]. In contrast to our findings, p53 seems to be critical for caspase-2-mediated apoptosis in HCT116 cells in response to DNA damage [61]. Although PIDDosome-independent actions by caspase-2 have been demonstrated by various studies, the functional impact of the PIDDosome in DNA damage is still controversially discussed. In vivo studies in mice indicate that the genetic loss of PIDDosome function does neither impair the DNA damage response nor affect the tumor suppressive capacity of caspase-2 presumably due to the compensation by PIDDosome-independent mechanisms [151]. In addition, previous data indicated that DNA damage triggers an ATM/ATR pathway which activates caspase-2-mediated apoptosis independent of p53, mainly through inhibition of the Chk1 [62].

### 3.3. Caspase-2 Acts as a Guardian of Genomic Stability

Genomic instability is a hallmark of many tumor cells and is observed in the majority of solid tumors. It can either result from structural lesions of the DNA, for example, truncations, translocations, or arise from chromosomal deletions. In addition, genomic instability can origin from alterations in chromosome numbers leading to aneuploidy which represents the most common chromosome abnormality in humans (for a review see [152,153]). As mentioned before, caspase-2-mediated apoptosis is critical for deleting mitotically aberrant and aneuploid cells as these defective cells have a high potential for tumorigenic transformation [67,68]. Since aneuploidy and chromosomal instability is a characteristic feature of caspase-2^−/−^ tumors in mice, apoptosis of aneuploidy cells is considered as a major tumor suppressive activity of caspase-2 [63,64]. Hence, caspase-2 inactivation or deficiency promotes aberrant DNA damage response and genomic instability leading to an enhanced clonogenic survival of aneuploid cells [63]. According to this study, MEFs from caspase-2^−/−^ mice are characterized by an increased resistance to apoptosis induced by microtubule and spindle poisons and increased DNA damage following irradiation [67]. Importantly, the caspase-2 dependent cleavage of Mdm2, described above, is part of a p53 regulatory pathway which is highly relevant for caspase-2 acting as a genomic stabilizer. A recent study that investigated the mechanisms which permit aneuploidy tolerance in colorectal cancer showed that caspase-2 is activated upstream of p53 in response to failed aggregation of chromosomes [68]. Subsequently, caspase-2 can avoid aneuploidy by two independent mechanisms. First, caspase-2 via cleavage of Mdm2 promotes generation of an Mdm2 fragment with a conserved p53 binding domain which is devoid of the RING domain critical for the ubiquitin ligase activity of Mdm2. Consequently, since p53 is no longer a target of proteasomal degradation, it accumulates, and this ultimately promotes cell cycle arrest and apoptosis upon genotoxic stress [144] (Figure 2B). Alternatively, activation of caspase-2 via cleavage of the BH3 interacting domain death agonist (Bid) which leads to an increase in truncated Bid (tBid) can promote apoptosis upstream of the mitochondrial outer membrane permeabilization (MOMP), caspase-3 and caspase-9, and independently of p53 [154] (Figure 2B). These findings are in accordance with our studies which demonstrated that siRNA-mediated knockdown of HuR via upregulation of caspase-2 further decreased clonogenic cell survival of colorectal carcinoma cells in response to γ-irradiation [87]. Similarly, caspase-2-triggered sensitizing effects on DNA-damage-induced apoptosis were observed upon treatment of colorectal cells with genotoxic drugs. Again, the caspase-2-dependent increase in apoptosis sensitivity and loss of clonogenic survival were observed in p53 wildtype RKO but also in p53 mutated DLD-1 cells, confirming that the sensitizing effects by caspase-2 occur independent of p53 [88]. By questioning the possible reasons for genomic instability frequently observed in many human tumors, a previous study revealed that loss-of-function alterations in the BCL9L are frequent in aneuploid colorectal tumors. Most intriguingly, dysfunction of BCL9L promotes a survival mechanism termed aneuploidy tolerance mainly through reducing the expression of caspase-2 (Figure 2B). Because caspase-2 is required for p53 stabilization and induction of aneuploidy-induced apoptosis, loss-of-function alterations in the tumor suppressor BCL9L seem to critically contribute to colorectal cancer. The authors support a model in which BCL9L constitutes an essential part of the β-catenin/transcription factor 4 (TCF4) complex which is critically involved in the activation of the caspase-2 gene promoter [68].

A former study demonstrated a RAS-triggered downregulation of procaspase-2 which is required for malignant transformation of intestinal epithelial cells [139]. Therefore, HuR may impair caspase-2-mediated deletion of aneuploid intestinal carcinoma cells by apoptosis in several respects; firstly, through stabilization of Mdm-2 encoding mRNA and secondly, via suppression of caspase-2 translation which again will reduce p53 levels but additionally cause a reduction in Bid-cleavage (Figure 2B).

## 4. Concluding Remarks

Many studies have shown that an overexpression of the ubiquitous mRNA binding protein HuR, due to its multifunctional roles on diverse mRNA functions, can play an essential role in tumorigenesis. In addition to the fact that many identified target genes of HuR account as key regulators of oncogenic pathways, studies from the last decade have emphasized the strong impact of HuR in the cell survival and DNA damage response of cells. A defect surveillance of these pathways is critical for the development and progression of many human malignancies, including colorectal cancer, and is closely related to therapy resistance of tumors. At first glance, the identification of caspase-2 as a novel target of HuR in colon carcinoma cells simply underscores the broad post-transcriptional cell survival program by HuR. However, with respect to several non-apoptotic functions which have been additionally assigned to this caspase family member including its functional role in genomic stability, DNA damage response, and cell cycle control, the inhibition of these tumor-suppressive functions may provide additional clues to the complex repertoire of cell survival and drug resistance mechanisms by HuR (Figure 3). The strong impact of caspase-2 on cell survival of colon carcinoma cells, which is underscored by the finding that the HuR-silencing-dependent sensitizing effects are almost completely rescued after genetic or pharmacological depletion of caspase-2, is unexpected since HuR can promote cell survival through targeting a broad panel of apoptosis-regulatory factors. On the basis of these pleiotropic functions in tumor development and progression, inhibition of HuR either alone or in combination with established therapeutic approaches offers an attractive novel strategy for treatment of some common human cancers including colorectal carcinoma. However, keeping in mind that HuR is indispensable for normal life and critical for epithelial and blood progenitor cell development, strategies aiming at global inhibition of HuR expression should be considered with caution. Instead, alternative strategies which would allow for only a temporally limited interference with some specific activities, for example (e.g., its suppressive effect on caspase-2 translation) may presumably represent a more desirable approach (Figure 1). In line with the fact that diverse kinases of the DDR can jointly influence diverse HuR functions, further experimental work is warranted to define the critical impact of different post-translational HuR modifications by these kinases for constitutive inhibition of caspase-2 translation in colon carcinoma cells. Finally, we anticipate that perturbations of signaling pathways controlling the negative HuR-caspase-2 axis via inhibition of the different tumor-suppressive activities by caspase-2 may have a strong pathophysiological impact for colon carcinogenesis and therapy resistance.

## Figures and Tables

**Figure 1 cells-08-00797-f001:**
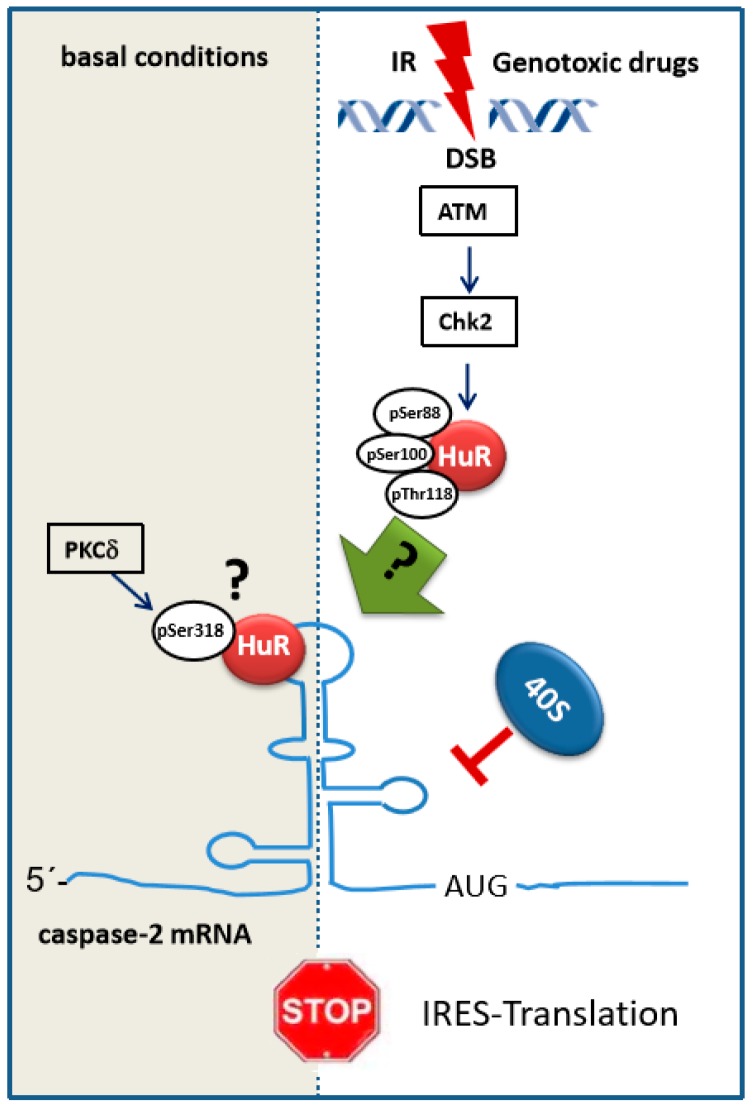
Schematic model of human antigen R (HuR) mediated inhibition of caspase-2 translation and its modulation by different kinase-triggered phosphorylation events. (white panel): DNA double-strand breaks (DSB) provoked by irradiation (IR) or genotoxic drugs via the activation of a canonical ATM-Chk2 axis can induce phosphorylation of HuR, and thereby may increase its binding affintiy to the IRES of caspase-2 mRNA (indicated by a green arrow). The depicted amino acid positions represent currently known Chk2-triggered HuR phosphorylation sites which are critical for the HuR-mRNA binding. In addition, the latent HuR binding to 5´UTR of caspase-2 mRNA which is observed in human colon carcinoma cells is presumably triggered by the constitutive HuR phosphorylation by PKCδ (grey panel). The question marks indicate that the final impact of different HuR phosphorylation events for IRES binding is currently not known. HuR may act as a negative ITAF which suppresses IRES-mediated translation of caspase-2 by preventing the recruitment of the 40S ribosomal subunit to caspase-2 mRNA. Activation of the DDR may further enhance the constitutive repression of caspase-2 translation as part of a so far unrecognized cell survival mechanism. Abbreviations: ATM, ataxia telangiectasia mutated; DSB, double-strand breaks; Chk2, checkpoint kinase 2; IR, irradiation; IRES, internal ribosomal entry site; PKC, protein kinase C.

**Figure 2 cells-08-00797-f002:**
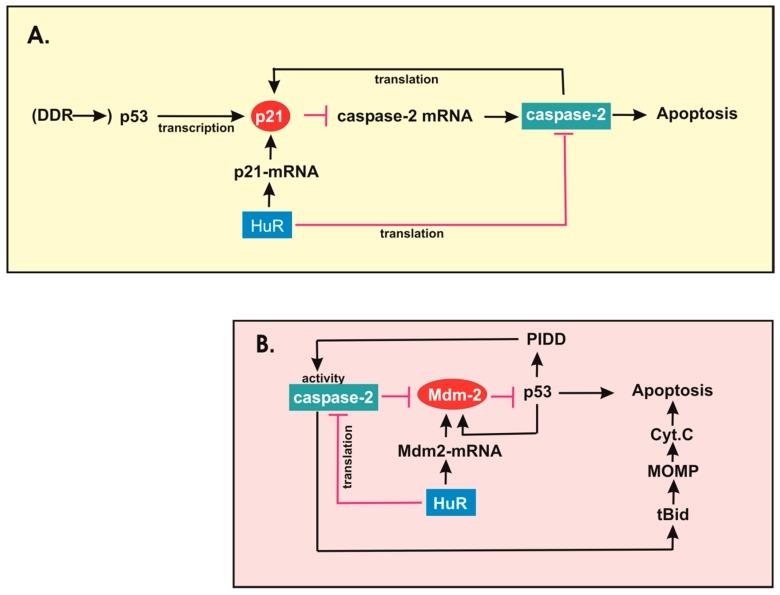
A complex interrelation of HuR and caspase-2 is implied in the control of apoptosis. (**A**) P53 activation either in response to DDR or independent of DNA damage (indicated by parenthesis), via the induction of p21 expression inhibits caspase-2 expression and consequently caspase-2-mediated apoptosis. HuR enhances this negative effect on caspase-2 through stabilizing p21-mRNA but, additionally, via direct inhibition of caspase-2 translation. On the contrary, caspase-2 can promote p21 translation by a mechanism which is independent of caspase-2 activity. (**B**) A similar antagonistic relationship between HuR and caspase-2 may play an important role in controlling Mdm-2. Arrows depict a positive and red lines a negative relationship. Abbreviations: Bid, BH3 interacting domain death agonist; DDR, DNA damage response; Cyt. C, cytochrome C; PIDD, p53-inducibleprotein with a death domain; MOMP, mitochondrial outer membrane permeabilization; tBid, truncated Bid.

**Figure 3 cells-08-00797-f003:**
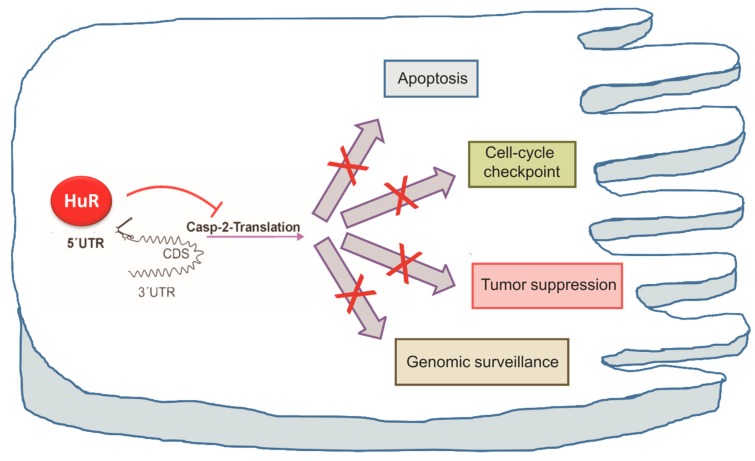
Inhibition of caspase-2 translation by HuR may affect different caspase-2-triggered cell functions. The HuR-mediated inhibition of these functions (indicated by red crosses) is presumably critically involved in the malignant phenotype and therapy resistance of colorectal cancer. Abbreviations: Casp2, caspase-2; CDS, coding sequence; UTR, untranslated region.

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
