# Peer review of "Inhibition of Caspase-2 Translation by the mRNA Binding Protein HuR: A Novel Path of Therapy Resistance in Colon Carcinoma Cells?"

_cells, 2019, doi:10.3390/cells8080797_

Reviewer 1 Report

Overall the manuscript by the Elberhardt et al is a well-researched and in-depth review of the regulatory pathways involving HuR and how it intersects with caspase 2 and impacts the DDR, apoptosis, cell survival and genomic stability and ultimately therapeutic resistance.

The main sections dealing with the role of HuR in carcinogenesis and the DNA damage response are appropriate and bring together many convergent as well as confounding data that reflect the multi-factorial nature of the regulatory mechanisms and signalling pathways explored.

The concluding remarks could be re-worked to remove the language errors and give more clarity and emphasis on the significant and new insights revealed by the review thereby providing a stronger finish.

The authors also need to be less subjective when referring to the work of their colleagues. I would suggest removing the word “excellent” in lines 70 and 328 as it is a personally biased comment and unnecessary.

The sentence in lines 517-518 doesn’t make sense and needs to be re-worded. The last sentence (lines 551-554) needs to begin with “Finally,” and “may” needs to be inserted after “caspase-2” (line 553) as the impact isn’t absolutely confirmed.

There are some issues around the grammar which needs to be addressed and can be fixed quite easily by global find and replace.

These include:

1.     Extensive usage of “implicate that” or “implicates that” or “implicating that” which I believe the authors actually means “imply that” or “suggest that”.

2.     The lack of hyphenation of prefixes in words such as posttranscriptional, preinitiation, posttranslational, nonmutually, interrelationship, preinitiation. The use of hyphens is especially helpful for readers when the same letters are being brought together.

3.     Replace “implicating” with “suggesting” on line 144 and in other places.

4.     Move the abbreviation DDR from line 265 to line 165 where DNA damage response is first used.

5.     Check “form” for where it is should to be “from”.

6.     Consider removal of unnecessary usage of “that" and “the” in the manuscript. For example, in line 41 “binds to the b-catenin thereby…”, and line 44 “analysis implicate that genetic mutations…”.

7.     “Consensually implicate” (line 413) is poor grammar and should to be replaced with something like “similarly demonstrate”.

8.     In line 407 “will needs to be replaced with “may” as this is presumed.

9.      In Figure 3 legend “cds” needs to be replaced by “CDS”.

10.  Where is Figure 3B as referred to in lines 456, 500, 512 and 522?

Finally, the authors need to run a close editorial eye over the whole manuscript to remove the minor typographic and grammatical errors present throughout but is particularly prevalent in the concluding remarks. For example, words such as “in virtue”, “exceptionally assigned”, “strong impact”, “aiming on” render some of the sentences where they are used much less understandable.  The first two sentences of the concluding remarks probably need to be re-written to remove the duplication when referring to the “last decade” and other less well chosen word usage.

Author Response

We would like to express our gratitude towards you for your overall positive evaluation and your valuable comments.

All points raised were taken up and the manuscript was revised in accordance with the reviewer´s suggestions. Any changes in the text were clearly highlighted by use of “Track changes” function in Microsoft Word. I would like to thank you for reconsidering the revised version of our manuscript for publication and hope for a positive decision

Reviewer 2 Report

This review summarized the role of RNA binding protein human antigen R (HuR) in colon carcinogenesis and the mechanisms underlying HuR’s modulatory activity on internal-ribosome entry site-triggered translation. In addition, this review describes the relationship between HuR and caspase-2 regulation, and the roles of the HuR-caspase-2 axis on cellular DNA damage response such as cell cycle arrest, apoptosis induction, and genomic stability. I think that this review is well-written and well-organized, and is important for the field of Cell biology. Furthermore, this review may give a useful information to the field of Oncology because the caspase-2 regulation by HuR and the therapy resistance of colorectal cancers is also discussed in this review. On the whole, in my opinion, this manuscript suitable for the publication in Cells.

Author Response

We would like to express our gratitude for the positive evaluation of our manuscript.